# Optimization Study on Synergistic System of Photocatalytic Degradation of AR 26 and UV-LED Heat Dissipation

**Chen Wang** [1] 🆔, **Haoliang Bai** [1] **and Xue Kang** [2,3,*]

1    School of Environment and Safety Engineering, North University of China, Taiyuan 030051, China
2    School of Chemistry and Chemical Engineering, North University of China, Taiyuan 030051, China
3    Dezhou Industrial Technology Research Institute of North University of China, Dezhou 253000, China
*    Correspondence: 20170032@nuc.edu.cn

**Abstract:** In this work, a novel UV-LED/TiO$_2$ photocatalytic system, having a single layer with ten LED beads, was designed to simultaneously achieve UV-LED cooling and wastewater degradation, to deal with heat dissipation problems of high-power UV-LEDs. To gain more insight into this system, the parameters affecting both cooling and photocatalytic performance were first optimized using AR 26 as a basis. With respect to sewage, sewage with a flow rate of 80 mL/min and a temperature of 20 °C helped to keep a lower temperature of UV-LED, which benefits the long-term operation stability of LED beads. For parameters affecting the photocatalytic performance only, the experiments showed that TiO$_2$ with moderate dosing (0.75 g/L) under strong acid conditions (pH = 2) helped to further improve photocatalytic activity when the initial concentration of AR 26 was 45 mg/L. Lastly, to illustrate the advantages of this novel system, the performance of the synergistic system was compared with a conventional photocatalytic reactor with respect to degradation performance, optical quantum efficiency, and energy consumption. The results showed that the degradation efficiency and light source utilization ratio of this coupled system were, respectively, 2.1 times and 1.5 times as much as those of a conventional reactor. As the unit power consumption of the synergistic system was only 0.18-fold more than that of a conventional reactor, our work suggests that this synergistic system with the advantage of LED lamp beads has a bright future in dealing with refractory organic pollutants of sewage.

**Keywords:** high-power UV-LED; heat dissipation performance; photocatalysis; wastewater; parameter optimization

## 1. Introduction

Environmental pollution has become a threat to the survival of life on earth. Wastewater pollution from various social industries is the leading cause among all effects that cause pollution. One of the primary sources of water pollution is printing and dyeing wastewater, which has a high concentration of pollutants and complex composition and represents hard-to-degrade industrial wastewater [1,2]. According to statistics, azo dyes account for 70% of synthetic dyes in wastewater [3]. As azo dyes contain one or more azo groups (-N=N-), this type of printing and dyeing wastewater is difficult to degrade [4].

Researchers have sought to develop different approaches to degrade azo dyes, including physical [5–7], biological [8,9], and electrochemical methods [10–13]. However, all these methods are non-destructive and only convert the pollutants from one form to another, thus causing secondary pollution. Therefore, photocatalysis, one of the advanced oxidation technologies, has entered the picture [14]. UV-LED as a UV light source solves the problems of traditional UV mercury lamps of low efficiency, unstable working power, long start-up time, short life span, and danger of mercury leakage, and is widely used for the photocatalytic treatment of wastewater [15–17]. However, the conversion efficiency of high-power UV-LEDs is limited, with more than 80% of the energy converted to heat

and the heat flow density exceeding 100 W/cm$^2$ [18,19]. The LED chip temperature will increase dramatically, which will have a significant negative impact on both the luminous efficiency and the lifetime. Therefore, effective heat dissipation becomes an important issue limiting the application of high-power UV-LEDs in photocatalysis. Many studies have been dedicated to this issue, including investigation of passive cooling methods, such as heat pipes [20], thermoelectric cooling techniques [21], and liquid immersion phase change cooling techniques [22]. However, passive cooling systems can only cool low-power LEDs. Active cooling is essential for high-power LEDs. Chen et al. [23] proposed an air-cooled microchannel heat sink and an air-cooled heat sink for high-power LED cooling. They pointed out that an air-cooled heat sink should be used in low-power LED chipsets, and a water-cooled microchannel heat sink should be used when the air-cooled heat sink cannot meet the needs of the chipset. Gatapova et al. [24] investigated liquid jet array cooling of 300 W nominal power LED luminaire modules. It was possible to control the substrate heat flux to 125 W/cm$^2$ and keep the module surface temperature below 70 °C. Sahu et al. [25] investigated the thermal characteristics of single-nozzle spray cooling on high-power LED modules. The results showed that the spray cooling design could keep the junction temperature of a nominal 300 W power LED below 95 °C. Seo et al. [26] used a ferromagnetic fluid to cool high-power LEDs and compared it with air and water. The results show that the ferromagnetic fluid had higher cooling performance and lower junction temperature compared to air and water. This resulted in a 67.5% and 66.1% improvement in the illumination of high-power LEDs, respectively. In general, active cooling is effective for high-power LEDs, and the system's cooling performance is significantly better than that of gas when the liquid is used as the cooling medium. For high-power UV-LED photocatalytic reactors, dyeing wastewater with high thermal conductivity and high specific heat capacity is an excellent cooling fluid at low cost. The wastewater will show a certain degree of temperature rise after absorbing the waste heat of a high-power LED, which is also beneficial to the photocatalytic reaction. To the best of our knowledge, few papers have reported coupling the cooling system of UV-LED arrays in photocatalytic reactors with wastewater degradation systems.

In one of our previous studies [27], a synergistic system was designed and proven to be useful for dealing both with the heat dissipation of UV-LED and the photocatalytic degradation of acidic red 26. However, as well as structural optimization [27], the optimization of the operating and reaction conditions should also be considered, which is one of the main objectives of this work. Hence, in the current work, the wastewater flow rate, wastewater temperature, initial pollutant concentration, catalyst concentration and pH were investigated. Moreover, using measures of the degradation performance, photometric efficiency and energy consumption, a comparison between the optimized system and the conventional system was also performed in this work.

## 2. Results and Discussion

For the UV-LED/TiO$_2$ synergistic system, the wastewater was circulated on the surface of the copper plate to cool the UV-LED, and the wastewater containing TiO$_2$ was degraded under UV-LED irradiation. The operating parameters affected both the UV-LED array cooling performance and the wastewater photocatalytic degradation performance. The wastewater flow rate and the inlet temperature influenced the cooling of the UV-LED and the photocatalytic performance. Other operating parameters, such as the initial concentration of reactants, catalyst concentration and pH, affected the photocatalytic reaction. Thus, the results were divided into two parts: (1) investigation of parameters affecting both the cooling and photocatalytic performance, and (2) exploration of parameters affecting the photocatalytic performance only.

### 2.1. Optimization of Parameters Affecting Cooling and Photocatalytic Performance

As a coolant, the flow rate and temperature of the wastewater affect the cooling performance of the UV-LED array, which in turn affects the luminescence efficiency of

LED beads, and, ultimately, the degradation performance of wastewater. In addition, as a reactant, the flow rate and temperature of the wastewater also directly affect the photocatalytic reaction, so the cooling performance and degradation performance need to be studied separately, as shown in Figure 1. Therefore, the cooling system was separated from the reaction system. A quantity of 140 mL of cooling water on the surface of the UV-LED array substrate was circulated by a peristaltic pump, and the inlet temperature was controlled by a thermostatic system. The 400 mL of wastewater at the bottom of the reactor was circulated by a peristaltic pump and the inlet temperature was controlled by a temperature control device.

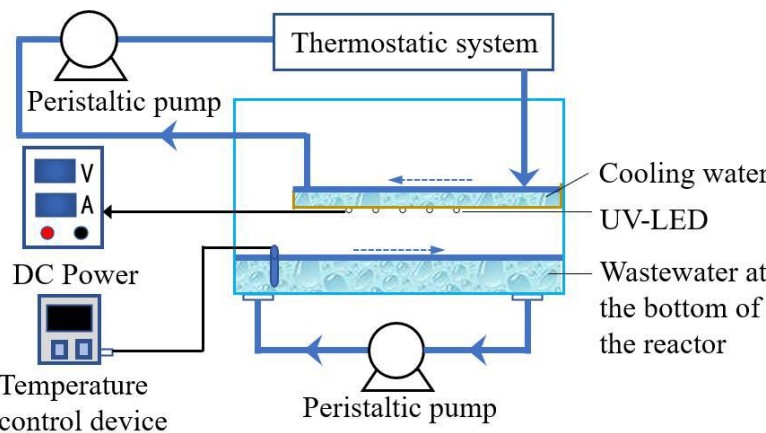

**Figure 1.** Schematic diagram of a degradation system that circulates cooling water and wastewater separately.

### 2.1.1. Flow Rate

In designing and applying a synergistic system, the flow rate plays a crucial role in degrading dyes [2,28]. The initial concentration of AR 26 was 15 mg/L; that of $TiO_2$ was 1 g/L; the cooling water and bottom wastewater temperature was 25 °C; and the wastewater flow rate was fixed at 80 mL/min. The cooling water flow rate on the substrate surface of UV-LED was adjusted to change in the range of 20~80 mL/min to investigate the effect of the cooling water flow rate on the temperature change rule of the UV-LED array; the results are shown in Table 1. The results show that the higher the flow rate, the better the cooling effect and the lower the temperature of the UV-LED array. When the cooling water flow rate increased from 20 mL/min to 80 mL/min, the average temperature of the UV-LED array decreased from 37.5 °C to 35.2 °C, and the junction temperature decreased by 6.1%. This was because the cooling water inlet temperature remained constant, and the faster the flow rate, the more cooling water passed through the unit copper plate area. According to the forced convection heat transfer mechanism, more heat was taken away at high flow rates. In addition, the lower cooling water flow rate caused a slight increase in water temperature. The temperature rise of the cooling water was 2.1 °C when the flow rate was 20 mL/min, which was seven times the temperature rise of the cooling water when the flow rate was 80 mL/min. Therefore, a higher flow rate favored a cooling effect, and lower flow rates caused a slight temperature increase enhancing the photocatalytic reaction rate.

**Table 1.** The cooling effect of the copper plate under different cooling water flow rates.

| Cooling Water Flow Rate (mL/min) | The Temperature at $t_1$ (°C) | The Temperature at $t_2$ (°C) | The Temperature at $t_3$ (°C) | The Temperature at $t_4$ (°C) | Average Temperature (°C) |
|---|---|---|---|---|---|
| 20 | 36.8 | 42.3 | 39.3 | 31.6 | 37.5 |
| 40 | 35.7 | 41.1 | 38.7 | 30.6 | 36.4 |
| 60 | 35.3 | 40.6 | 37.7 | 30.3 | 36.0 |
| 80 | 34.5 | 40.2 | 36.1 | 29.9 | 35.2 |

The wastewater flow rate affected the residence time and mass transfer of wastewater in the synergistic system. The initial concentration of AR 26 was 15 mg/L, the concentration of $TiO_2$ was 1 g/L, the temperature of the cooling water and the bottom wastewater was 25 °C, and the flow rate of the cooling water was 80 mL/min. The degradation experiment was carried out by adjusting the wastewater flow rate in the range of 20~80 mL/min. The results are shown in Figure 2. A higher cooling water flow rate offered better degradation performance until the wastewater was decolorized. This was because the wastewater retention time in the pipeline decreased with increase in flow rate, which meant the cumulative UV irradiation time rose. When the flow rate reached 80 mL/min, the overall exposure time of the wastewater to UV irradiation was the longest, and the dye decolorization rate was the highest at 97.3%. However, at the lowest flow rate, 20 mL/min, the residence time in the pipeline was the longest. The exposure time to UV irradiation of the whole wastewater was the shortest, so the dye decolorization rate of 94.6% was lower than that at the highest flow rate. Thus, the wastewater flow rate was adjusted to 80 mL/min to achieve a better cooling effect and decolorization performance.

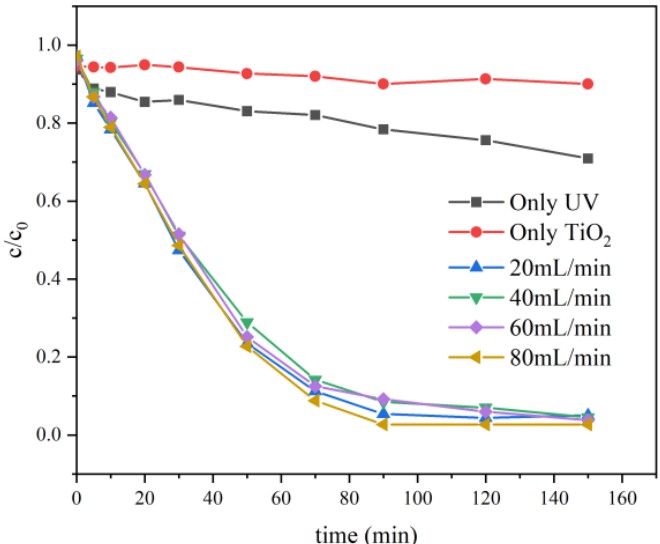

**Figure 2.** Concentration variation of AR 26 at sewerage flow rates of 20, 40, 60, and 80 mL/min.

### 2.1.2. Wastewater Temperature

Wastewater circulates on the surface of UV-LED arrays and takes away the waste heat, thus affecting the UV-LED light irradiation and wastewater degradation. The wastewater temperature will also affect the degradation performance. To investigate the effect of the wastewater temperature on both the cooling and photocatalytic reaction performance, the temperature of the cooling water on the UV-LED copper board and the temperature of the wastewater at the bottom of the reactor were controlled separately.

### Variation of Cooling Water Temperature on UV-LED Array

With a constant wastewater temperature of 25 °C, only the temperature of the UV-LED cooling water was changed at 0~80 °C. The results are shown in Figure 3. As the cooling water temperature increased from 0 °C to 80 °C, the average temperature of the LED substrate increased from 16.8 °C to 78.5 °C, and the junction temperature rose from 30 °C to 91.7 °C.

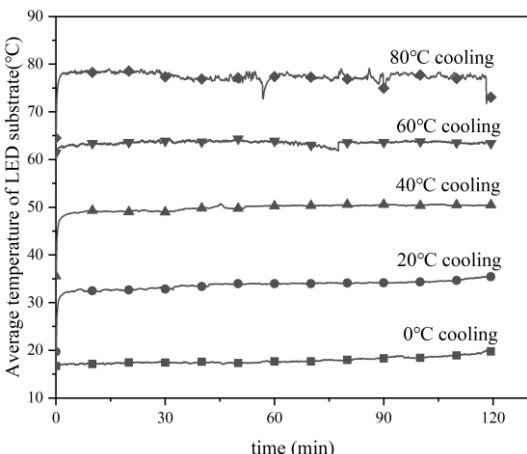

**Figure 3.** The average temperature of the LED array substrate at cooling water temperatures of 0, 20, 40, 60, and 80 °C.

In addition, the effect of the wastewater temperature on UV-LEDs was also reflected in the electro-optical conversion efficiency. Therefore, a light power meter probe was set up 20 mm below the UV-LED array (i.e., at the bottom wastewater surface), and the irradiance of the six sampling points was tested. The results are shown in Figure 4. As the cooling water temperature increased from 0 °C to 80 °C, the average irradiance decreased by 17.2%. Due to the increase in the junction temperature, the lattice vibration amplitude of the semiconductor was enhanced. The chance of a radiation-free electron jumping in the PN junction was significantly increased, making the internal quantum efficiency decrease. This eventually led to a decrease in the irradiance [29].

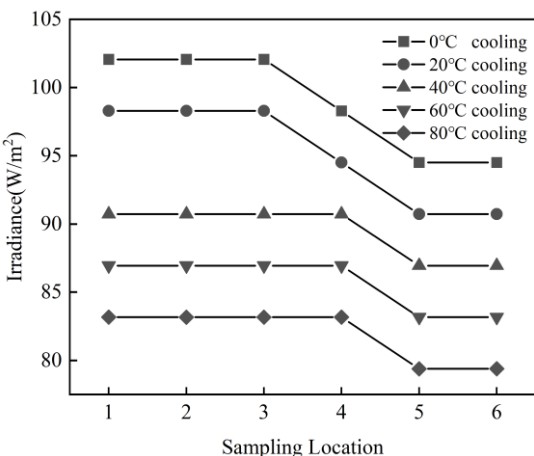

**Figure 4.** Irradiance of the UV-LED array at 6 positions under cooling water temperatures is 0, 20, 40, 60, and 80 °C, respectively.

In addition, lower light intensities were observed at positions 5 and 6. This was due to the flow of coolant from positions 3 and 4 to positions 5 and 6 (as shown in experiemental parts), which created a temperature gradient. The UV-LED beads at positions 5 and 6 operated at higher temperatures for longer periods of time, which could cause irreversible damage to the internal structure. This led to a decrease in the luminous efficiency of the lamp beads at this location.

The effect of the wastewater temperature on UV-LED arrays is ultimately reflected in an impact on degradation performance. The effect of the cooling water in the temperature range of 0~80 °C on the degradation of AR 26 is shown in Figure 5. The results illustrate that the impact of the UV-LED array working temperature on the degradation performance was insignificant in the cooling water temperature range of 0 to 60 °C. However, as the cooling

water temperature increased from 60 °C to 80 °C, the AR 26 reaction rate constant decreased from 0.045 to 0.027, decreasing by 40%. The reason for the drastic decrease in reaction rate was the reduction in the electro-optical conversion efficiency of the UV-LED array at high temperatures, which coincides with our results for the irradiance tests (Figure 4).

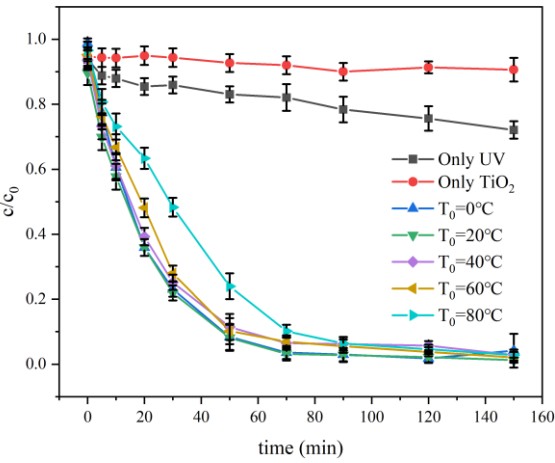

**Figure 5.** Variation of AR 26 concentration at cooling water temperatures of 0, 20, 40, 60, and 80 °C.

Variation of the Temperature of Wastewater at Bottom of Reactor

To study the effect of the wastewater temperature on the photocatalytic reaction, the temperature of the wastewater at the bottom of the reactor, ranging from 0 °C to 80 °C at a constant cooling water temperature of the UV-LED array (25 °C), was controlled. The experimental results are shown in Figure 6. It can be seen that the reaction rate increased as the temperature of the wastewater at the bottom of the reactor rose. When the temperature was low (0~60 °C), the temperature of the wastewater was proportional to the degradation rate. The reaction rate constant of 60 °C wastewater (k = 0.10) was five times higher than that of 0 °C wastewater (k = 0.02). However, when the temperature of the wastewater at the bottom of the reactor increased to 80 °C, it was observed that the reaction rate increased by only 2.4% and the degradation rate tended to a maximum.

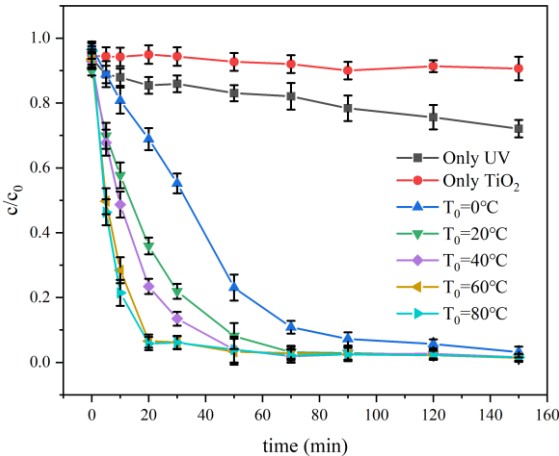

**Figure 6.** Evolution of the concentration of AR 26 at temperatures of wastewater at the bottom of the reactor of 0, 20, 40, 60, and 80 °C.

Regarding the effect of temperature on the rate of photocatalytic reaction, this can be explained as follows: ① Adsorption of organic matter: adsorption, as an exothermic reaction, readily occurs at lower temperatures (<80 °C) and does not readily occur at higher temperatures; ② Apparent activation energy: Due to the activation effect of photons, the

apparent activation energy of the photocatalytic reaction is usually the smallest at 20~80 °C, and the reaction can occur easily. At low temperatures (<0 °C), the apparent activation energy increases, making the photocatalytic reaction challenging [30]; ③ Charge-hole complexation: It has been shown that overly high reaction temperatures (>80 °C) promote charge-hole complexation, which is not conducive to the adsorption of organic compounds on the catalyst surface [31].

Therefore, the low performance at shallow temperatures (0 °C) can be explained by increase in the apparent activation energy and the fact that the thermal desorption of the products becomes exceptionally difficult at this time, which represents a rate-limiting step of the reaction. At extremely high reaction temperatures (80 °C), the exothermic adsorption of the reactants becomes unfavorable. This tends to be the rate-limiting step of the reaction, and the recombination of photogenerated electrons and holes further limits the photocatalytic reaction. Thus, the optimum temperature for photocatalytic reactions is between 20 and 60 °C.

To clarify the influence of single or multiple factors on performance and to obtain optimal conditions, statistical analysis of the results can be performed [32]. Matlab software was used to numerically determine the relationship between the first-order rate constant and the cooling water and the temperature of the wastewater at the bottom of the reactor, as shown in Figure 7. It can be seen that the effect of the temperature of wastewater at the bottom of the reactor and the LED array temperature on the reaction rate is opposite, and that the effect of the temperature of the wastewater at the bottom of the reactor is more marked. According to the above analysis, the suitable wastewater temperature is 20~60 °C. Although higher wastewater temperature results in a faster reaction rate, our previous studies have shown that an overly high wastewater temperature negatively impacts the long-term stability of LED beads [27]. Therefore, considering the long-term stability of operation, the optimal wastewater temperature is 20 °C (at room temperature).

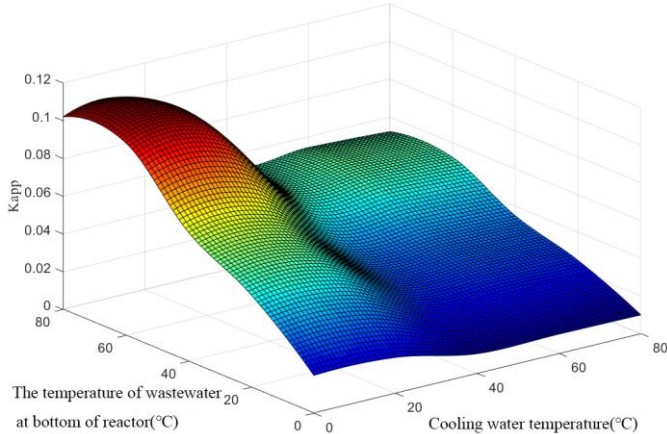

**Figure 7.** The numerical fit of first-order rate constants versus LED arrays and wastewater temperature.

### 2.2. The Effect of Photocatalytic Performance

#### 2.2.1. Initial AR 26 Concentration

To explore the effect of the initial concentration of AR 26 on photocatalytic degradation, wastewater with different AR 26 amounts (ranging from 15 to 60 mg/L) was prepared with a constant catalyst concentration (1 g/L). The wastewater temperature was kept at 20 °C. As shown in Figure 8, the photocatalytic degradation reaction of AR 26 accords with a pseudo-first-grade reaction kinetics model. The degradation rate decreased with increase in the initial concentration of the dye, and the time taken for degradation gradually increased. The rate constant of degradation decreased from 0.09 to 0.01, decreasing by 72.1%, when the initial concentration of AR 26 was increased from 15 mg/L to 45 mg/L. When the initial concentration of AR 26 exceeded 45 mg/L, the photocatalytic reaction proceeded slowly, and the degradation rate tended to level off. The degradation rate of the dye decreased with

increasing dye concentration. The degradation rate is related to the probability of formation of ●OH radicals on the catalyst surface and the probability of reaction of ●OH radicals with the dye molecules. This is because, at higher dye concentrations, the active sites are covered by dye ions, resulting in less ●OH radicals generated on the catalyst surface. In addition, at higher dye concentrations, the wastewater has a lower transmittance, and a large amount of UV light is absorbed by the dye resulting in fewer photons reaching the catalyst surface, which leads to a lower concentration of radicals, such as ●OH and $O^{2-}$, significantly reducing the photocatalytic activity.

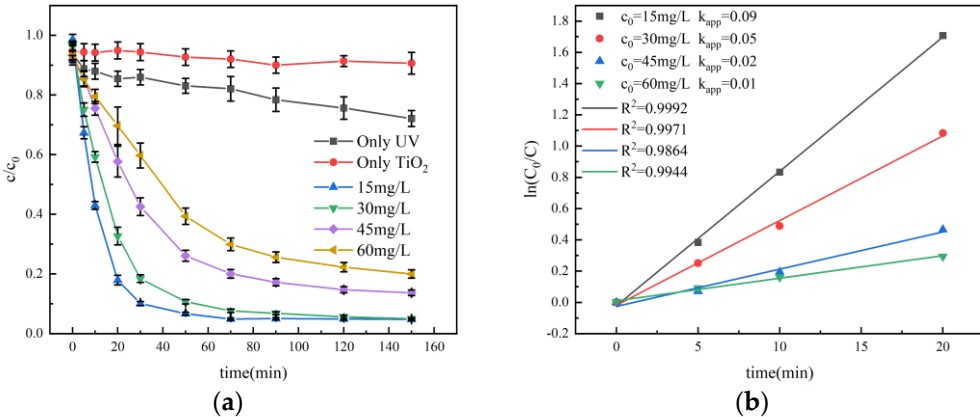

**Figure 8.** (**a**) Evolution of AR 26 concentration with different initial concentrations; (**b**) Kinetic fits for initial concentrations of 15, 30, 45, and 60 mg/L.

In addition, it was observed that the decolorization rate of AR 26 after 150 min also decreased with the initial concentration increasing. When the initial concentration of AR 26 increased from 15 mg/L to 45 mg/L, the degradation rate of AR 26 decreased from 95.18% to 78.61% after 150 min. This may have been because of the increased generated intermediates with time at higher dye initial concentration, which resulted in competition with the native pollutants for free radicals, such as ●OH and $O^{2-}$, and, finally, reached an equilibrium state. This is the main reason why the decrease in pollutant concentration tended to level off at the later stage of degradation. A similar conclusion was obtained by Petrucci et al. [13].

### 2.2.2. TiO$_2$ Dosage

To determine the optimal catalyst dosage for the photocatalytic reaction, 0~2 g/L TiO$_2$ was selected. As shown in Figure 9, the reaction rate constant increased from 0.002 to 0.014, and the degradation rate increased from 22.46% to 63.44%, with a TiO$_2$ dosage range of 0~0.5 g/L. As the catalyst concentration increased to 0.75 g/L, the reaction rate increased from 0.014 to 0.027, and the degradation rate increased from 63.44% to 84.20%. Increasing catalyst concentration was not able to significantly enhance the photocatalytic reaction rate and the reaction rate plateaued.

Among the reasons for the elevated degradation rate are the fact that the total surface area of the photocatalyst, i.e., the number of active sites for the photocatalytic reaction, increased with increase in the number of photocatalysts. However, there was no significant increase in the degradation rate with further increase in catalyst concentration (>0.75 g/L), which is consistent with the results reported in most studies [33,34]. As the catalyst concentration increases, the number of available active sites on the TiO$_2$ surface tends to be constant. Too high catalyst concentration increases the turbidity and light scattering of the wastewater, resulting in fewer photons reaching the catalyst surface and limiting further increase in the degradation rate [35]. In addition, the decrease in degradation rate at higher catalyst concentrations may be due to the deactivation of activated molecules by collision with ground state molecules [36]. It was also observed that, at higher catalyst concentrations (>2 g/L), agglomeration and deposition of TiO$_2$ particles occurred [37]. At

this point, a portion of the catalyst surface may not be available for photon absorption and dye adsorption, providing little stimulation to the catalytic reaction.

Therefore, the reaction should be operated below the saturation level of the $TiO_2$ photocatalyst to avoid catalyst overload and ensure efficient photon absorption. In this study, the optimal catalyst concentration was 0.75 g/L to maximize degradation efficiency.

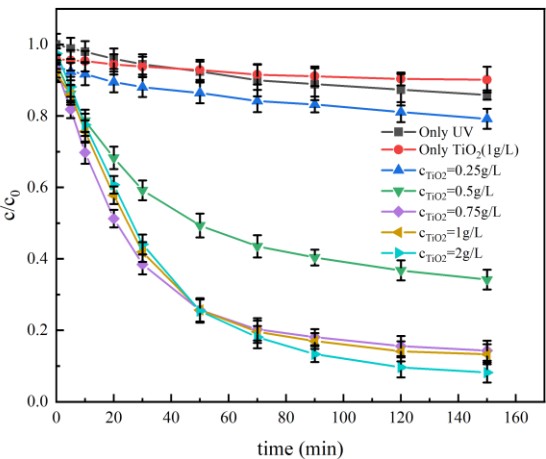

**Figure 9.** Evolution of concentration with four $TiO_2$ concentrations at a wastewater temperature of 20 °C and AR 26 initial concentration of 45 mg/L.

### 2.2.3. pH

pH affects the aggregation of semiconductor catalyst particles, the catalyst surface's electrical properties, and the organic matter adsorption on the catalyst surface. The effect of pH on the photocatalytic degradation rate was investigated by adding appropriate amounts of sodium hydroxide or dilute hydrochloric acid to adjust the pH between 2 and 8. The initial pH of the wastewater was 6.4. The results are shown in Figure 10. When the pH was reduced from 6.4 to 4.0, $k_{app}$ was reduced from 0.016 to 0.011 and the degradation rate was reduced by 20.21%. When the pH was further reduced to 2.0, the $k_{app}$ increased from 0.011 to 0.021, and the degradation rate increased by 38.34%. However, when the pH increased from 6.4 to 8.0, the degradation rate decreased from 0.016 to 0.003, decreasing by 78.79%.

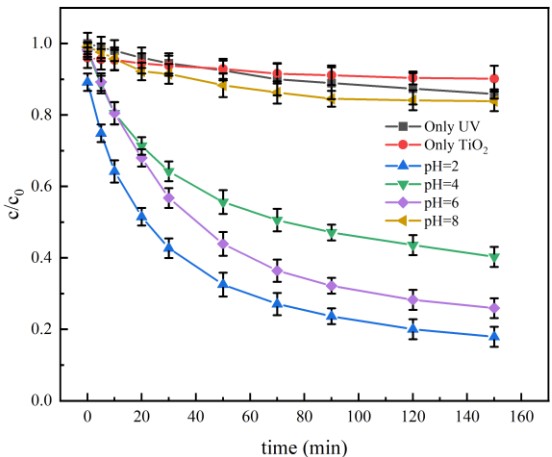

**Figure 10.** Evolution of AR 26 photocatalytic degradation in the pH range of 2 to 10 with time at a wastewater temperature of 20 °C, an initial AR 26 concentration of 45 mg/L, and a $TiO_2$ concentration of 0.75 g/L.

The pH will affect the adsorbed state and the oxidated group, leading to different photocatalytic reaction rates. As the isoelectric point of P25-type $TiO_2$ is 6.9 [38], the $TiO_2$

surface is positively charged when pH < 6.9. This will lead to more anionic dyes adsorbed on the $TiO_2$ surface, which is favorable for the photocatalytic reaction. In addition, the ratio of photogenerated holes to hydroxyl radicals is influenced by the pH value, which also affects the photocatalytic reaction [39].

As shown in Figure 10, as the pH decreased from 6 to 4, although the adsorption ability and the relative content of the photogenerated cavities increased, the reduction in hydroxyl radicals was the main influencing factor at this time, leading to inferior reaction activity. When the pH was further reduced to 2, the enhanced adsorption and increase in the relative content of the photogenerated cavities became the main influencing factors, showing a superior reaction rate. This result suggests that the degradation of AR 26 is sensitive to the pH.

However, when the pH was higher than the $TiO_2$ isoelectric point, the less adsorbed dye species played a significant role in determining the photocatalytic activity. Here, pH = 2 was chosen as the optimal condition for the degradation of AR 26.

### 2.3. Performance Comparison with Conventional Photocatalytic Reactor

#### 2.3.1. Wastewater Degradation Performance

The two photocatalytic system degradation experiments were performed with a wastewater flow rate of 80 mL/min, a temperature of 20 °C, an AR 26 concentration of 45 mg/L, a $TiO_2$ concentration of 0.75 g/L, pH = 2, and a wastewater volume of 400 mL. The results of comparing the decolorization effect of their wastewater are shown in Figure 11a. The results showed that the degradation rate of AR 26 in the conventional photocatalytic reactor was about 73.56% at 150 min, while the coupled system achieved a 72.77% degradation rate of AR 26 at 70 min. The two degradation reactions were fitted kinetically, and the comparison results are shown in Figure 11b. The photocatalytic reaction rate constant of the coupled system was 2.1 times as much as that of the conventional photocatalytic reactor. Moreover, the coupled system had a low-power light source (12 W vs. 32 W) and a long wavelength of UV beads (380 nm vs. 365 nm). However, an irradiation flux of 1.62 W was much higher than the conventional photocatalytic reactor of 0.2 W, which led to more efficient effluent degradation and illustrated the coupled system's structural superiority.

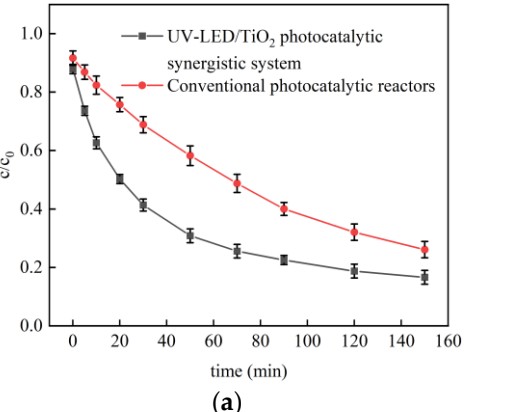
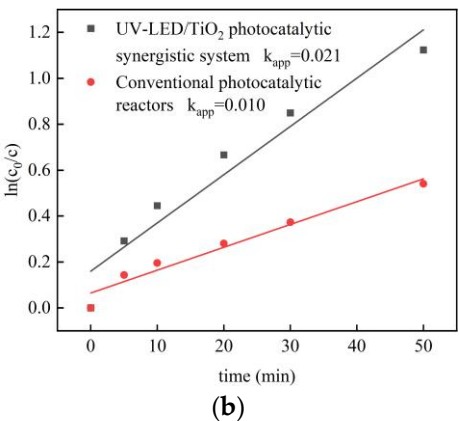

(**a**)    (**b**)

**Figure 11.** (**a**) Comparison of wastewater degradation performance between UV-LED/$TiO_2$ photocatalytic synergistic system and conventional photocatalytic reactor when the wastewater flow rate was 80 mL/min, the temperature was 20 °C, initial AR 26 concentration was 45 mg/L, $TiO_2$ concentration was 0.75 g/L and pH = 2; (**b**) Kinetic fit for two photocatalytic systems.

On the other hand, it is seen from the 2.1 and 2.2 results that temperature positively affected the removal of AR 26. The temperature rise due to the absorption of heat from the UV-LED array by the effluent in the coupled system also had a positive effect on the degradation performance. Ennouri et al. [40] also obtained similar findings. This suggests that the decolorization and degradation performance of the coupled system for AR 26 is better than that of the conventional photocatalytic reactor.

### 2.3.2. Optical Quantum Efficiency

Since the light source types of the two photocatalytic systems are different, it is more appropriate to evaluate their performance using the apparent quantum yield AQY.

The irradiance $I$ of the coupled system and the conventional photocatalytic reactor measured by Equation (3) were 95.13 W/m$^2$ and 487.26 W/m$^2$, respectively. The incident light wavelengths $\lambda$ of the two light sources provided by the vendor were approximately 380 nm and 365 nm. The incident light areas $A_L$ of the two reactors were calculated to be $1.71 \times 10^{-2}$ m$^2$ and $3.90 \times 10^{-3}$ m$^2$, respectively. The degradation rates $v$ were $8.73 \times 10^{-9}$ mol/s and $6.38 \times 10^{-9}$ mol/s. The AQY of the coupled system was calculated to be 1.5 times that of a conventional photocatalytic reactor, proving that the AR 26 degradation reaction in the coupled system had higher light quantum utilization for UV-LED.

### 2.3.3. Analysis of Energy Consumption

The photocatalytic degradation process's energy consumption is a critical indicator for evaluating the performance of photocatalytic reactors.

This section compares the energy consumption of the coupled system with that of the conventional ring-gap photocatalytic reactor by calculating the unit electrical energy consumption. The power of the light source of the coupling system and the traditional photocatalytic reactor was 12 W and 32 W. At 50 min, the concentration of AR 26 was 14.63 mg/L and 0.23 mg/L, respectively. According to Equation (8), the EE/O of the two can be calculated as 193.90 kW·h and 1075.13 kW·h. The energy consumption of the coupled system was only 18% that of the conventional ring-gap reactor. In the work of Quan et al. [41], it was found that the EE/O of the gas–liquid–solid cycle slurry photocatalytic and the ring-gap photocatalytic reactor were 358 kW·h and 218 kW·h, respectively. Even compared to their results, the newly coupled system showed obvious superiority with respect to energy-saving.

## 3. Materials and Methods

### 3.1. Materials

The experimental photocatalyst was P25 titanium dioxide with an average particle size of 30 nm and a specific surface area of 50 m$^2$/g (Fuchen Chemical Reagent Co., Ltd., Tianjin, China). The particles' mass ratio of anatase to rutile was 4:1. Acid Red 26 (AR 26, CAS No. 3761-53-3, Ruji Biotechnology, Shanghai, China).

A high-power UV-LED (Model: 3535 violet light bead, Taiwan Togia Optoelectronics Technology Co., Ltd., Taiwan, China), with lamp bead diameter of 3.45 mm, rated power of 3 W, luminous wavelength of 380~385 nm, and working temperature of 30~80 °C was used. The irradiance of the UV-LED beads was tested with a optical power meter (Model: PL-MW2000, Beijing Porphyry Technology Co., Ltd., Beijing, China).

### 3.2. UV-LED/TiO$_2$ Photocatalytic Synergistic System

The UV-LED/TiO$_2$ photocatalytic synergistic system comprises a photocatalytic reactor, a wastewater circulation module, and a data acquisition system. As shown in Figure 12a, the photocatalytic reactor is a glass vessel (190 × 90 × 190 mm) with a luminescence system, which presents 10 UV-LED beads with single-layer copper plates (170 × 90 × 2 mm). Ten UV-LED beads were evenly arranged at the bottom of the copper plate, with five beads in each group connected in series and two groups merged in parallel, as shown in Figure 12b. The UV-LED beads were glued to the bottom of the copper plate by thermal conductive paste, and K-type thermocouples were fixed on the surface of the LED substrate, as shown in Figure 12c. The copper plate with UV-LED beads was set inside the reactor with the beads 20 mm from the wastewater surface in the vertical direction.

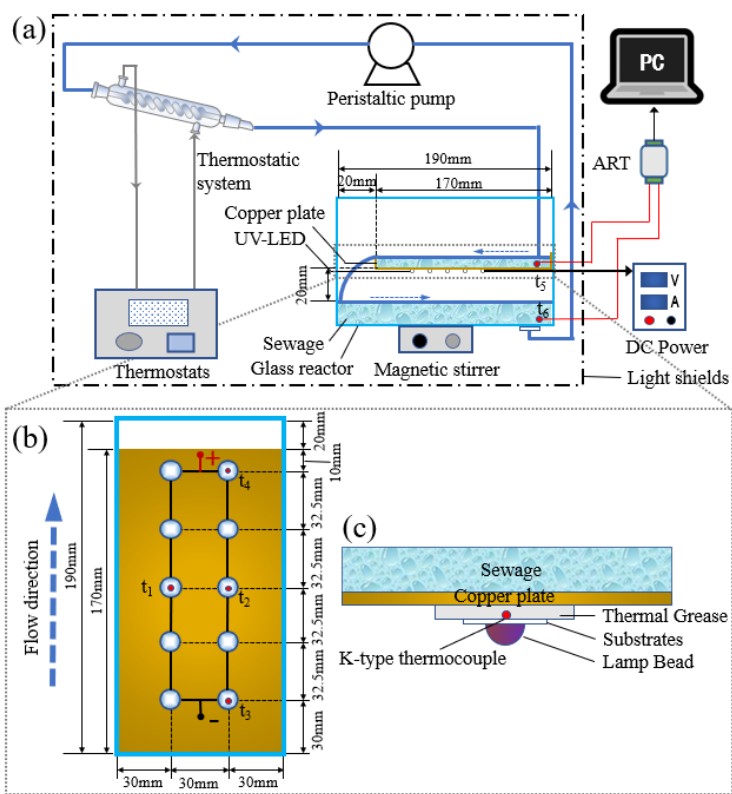

**Figure 12.** (**a**) Diagram of UV-LED/TiO$_2$ photocatalytic synergistic system. (**b**) UV-LED array layout diagram. (**c**) Vertical cross-section of lamp beads and copper plate.

The wastewater was circulated by a pump (Model: NKP-DA-S10B, Kamoer Fluid Technology Co., Ltd, Shanghai, China). It entered the reactor from the top inlet of the synergistic system. It flowed through the copper plate while carrying away the heat from the LED arrays. Upon reaching the bottom of the reactor, the wastewater underwent a photocatalytic reaction under UV light from the LED array. It was then pumped out and passed through the thermostat into the synergistic system again. During this process, the temperature distribution of the UV-LED array ($t_1 \sim t_4$), the temperature of the UV-LED substrate, and the temperature of the wastewater inlet ($t_5$) and outlet ($t_6$) were obtained by the K-type thermocouple (Model: TT-K-30, Omega Engineering Inc., State of Connecticut, Norwalk, CT, USA) and recorded in the ART temperature acquisition module.

### 3.3. Experimental Procedure

#### 3.3.1. Experimental Process of a Synergistic System

Wastewater with different AR 26 initial concentrations, catalyst amount, and pH was prepared. Before starting the illumination of the UV-LED beads, the wastewater containing the catalyst was stirred in the dark for 60 min until it reached adsorption equilibrium. The speed of the magnetic stirrer was adjusted to 180 r/min. The pump was turned on, the water began circulating, and the flow rate of the copper plate surface was about 17 cm/min. Then, the UV-LED beads were powered by a constant operating current of 0.69 A. The reactor performance was evaluated by taking samples at different time points and measuring the reduction in AR 26 concentration in the wastewater. The sampling times of the wastewater sample were set as: 0, 5, 10, 20, 30, 50, 70, 90, 120, 150, and 180 (min). After the experiment, the wastewater samples were centrifuged at 4000 r/s for 15 min, the supernatant was passed through a 0.22 um cellulose membrane for secondary filtration, and the AR 26 concentration was measured using a UV-Vis spectrophotometer. The temperature of the UV-LED array and the temperature of the wastewater inlet ($t_5$) and outlet ($t_6$) were recorded continuously.

3.3.2. Experimental Flow of Conventional Photocatalytic Reactor

The conventional photocatalytic reactor experiment was conducted using a photocatalytic reactor (Model: JH-HX-MC, Shanghai Jiheng Industrial Co., Ltd, Shanghai, China) with mercury lamp as the light source to simulate the degradation experiment of conventional ring-gap photocatalytic reactor. A compact arrangement of quartz tubes around the light source simulated the wastewater distribution in the ring-gap photocatalytic reactor. As shown in Figure 13, the radius of UV radiation was about 130 mm. A single quartz tube has a radius of about 12 mm and a volume of 50 mL. Theoretically, the number of quartz tubes required to fully utilize the UV of the mercury lamp should be 23. Therefore, the power of a single quartz tube was calculated to be 32 W. To compare with the performance of the coupled system in this paper, one quartz tube was used to degrade the same volume of wastewater (400 mL).

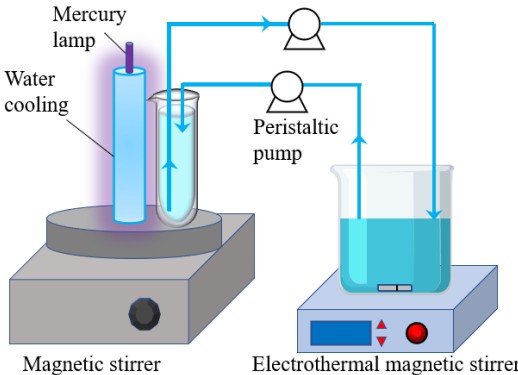

**Figure 13.** Schematic diagram of the experimental setup of a conventional photocatalytic reactor.

*3.4. Analysis Methods*

3.4.1. Heat Dissipation Performance of UV-LED

The heat transfer performance of the wastewater cooling system was evaluated by UV-LED substrate temperature, junction temperature, and luminous irradiance.

Four K-type thermocouples measured the temperature of the UV-LED substrate ($T_i$), and their average value was calculated to obtain the average temperature of the UV-LED array ($T_{UV\text{-}LED}$), which was calculated as follows:

$$T_{UV-LED} = \frac{1}{4}\sum_{i=1}^{4} T_i \tag{1}$$

UV-LED lamp beads of the substrate thermal resistance ($R_{substrate}$) containing 1.5 mm aluminum plate thermal resistance and the maximum thickness of 200 mm thin electrical insulation layer thermal resistance, usually to 0.5 °C/W, were suggested by the UV-LED supplier to give a PN junction to the solder joint thermal resistance $R_{j\text{-}sp}$ of about 6 °C/W. The UV-LED junction temperature ($T_{junction}$) can be calculated as Equation (2) [42]:

$$T_{junction} = T_{substrate} + R_{substrate} \times P_{array} + R_{j-sp} \times P_{single} \tag{2}$$

where $T_{substrate}$, $P_{array}$, and $P_{single}$ are the substrate temperature of the UV-LED beads, the total power of the UV-LED dense array, and the power of a single UV-LED bead.

The actual optical power ($I_i$) of the UV-LED array was measured with an optical power meter, as shown in Figure 14a. The optical power meter probe was placed at the same height as the wastewater water surface (20 mm), and the six measurement points were arranged as shown in Figure 14b. The irradiance ($E_i$) is the radiation flux per unit area of the irradiated surface, calculated as follows:

$$E_i = \frac{I_i}{A} \tag{3}$$

where $A$ is the light power meter probe receiving an irradiation area of 3.14 cm$^2$.

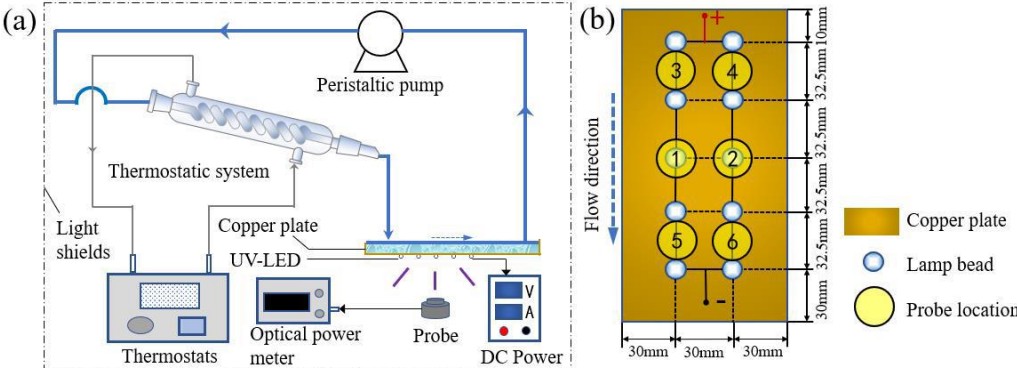

**Figure 14.** (**a**) Schematic diagram of the optical power measurement system. (**b**) Optical power meter probe location schematic. The six probe locations are numbered 1 to 6. The wastewater flow direction is from location 3, 4 to location 5, 6.

### 3.4.2. Photocatalytic Degradation Performance

The discoloration of AR 26 is related to the cleavage of the azo bond in the dye molecule. Azo dyes are characterized by an azo double bond (N=N) attached to two radicals, usually aromatic groups (benzene or naphthalene rings). The color of azo dyes is determined by the azo bond and its associated chromophores and auxiliary chromophores. These chromophores are the most active bonds in the azo dye molecule and can be oxidized by cationic vacancies or hydroxyl radicals or reduced by electrons in the conduction band. Therefore, dye decolorization implies breakage of the N=N bond.

To analyze the wastewater degradation performance, in addition to wastewater color change and concentration variation ($c/c_0$), the degradation performance can be evaluated by reaction kinetics to illustrate the degradation process. The L–H (Langmuir–Hinshelwood) kinetic model to describe the degradation kinetics of AR 26:

$$-\frac{dc}{dt} = \frac{k_{\mathrm{r}}kc}{1+kc} \tag{4}$$

where $c$, $t$, $k_{\mathrm{r}}$, $k$ are the AR 26 concentration, reaction time, adsorption constant, and reaction rate constant. $c_0$ is the initial concentration of AR 26.

The $k_{\mathrm{r}}$ and $k$ are mainly determined by the type and concentration of catalyst in the reaction system, the pH, the initial concentration of pollutants, the irradiance and the wastewater temperature, etc. These parameters are constant in this experimental system, and $kc << 1$, so Equation (4) is simplified to Equation (5):

$$-\frac{dc}{dt} = k_{\mathrm{app}}c \tag{5}$$

where $k_{\mathrm{app}}$ is the degradation rate constant.

An important indicator to evaluate the performance of the photocatalytic reactor is the apparent quantum yield (AQY); the number of incident photons is measured by a photo-power meter and calculated as follows [43]:

$$\mathrm{AQY} = \frac{N_{\mathrm{e}}}{N_{\mathrm{P}}} \times 100\% = \frac{v \times N_{\mathrm{A}} \times N}{I \times A_{\mathrm{L}}/E_{\lambda}} \times 100\% \tag{6}$$

where $N_{\mathrm{e}}$ is the number of electrons used for AR 26 degradation; $N_{\mathrm{P}}$ is the number of incident photons; $N$ is the number of electrons transferred by the reaction; $v$ is the degradation rate, mol/s; $N_{\mathrm{A}}$ is Avogadro's constant, $6.02 \times 10^{23}$ mol$^{-1}$; $I$ is the optical power density, W/m$^2$; $A_{\mathrm{L}}$ is the incident light area, m$^2$; and $E_{\lambda}$ is the energy of a photon at a specific wavelength, which can be calculated as follows:

$$E_{\lambda} = \frac{h \times f}{\lambda} \tag{7}$$

where $h$ is Planck's constant, $6.63 \times 10^{-34}$ J·s; $f$ is the speed of light, $3 \times 10^8$ m/s; and $\lambda$ is the wavelength of incident light, nm.

In addition, the unit electrical energy consumption (EE/O) is also an important indicator for evaluating the economics of photocatalytic reactors. It is defined as the amount of electrical energy (kW·h) required to reduce the concentration of pollutants in 1000 gallons (1 gallon = 3.785 L) of wastewater by one order of magnitude. The EE/O value can be calculated from Equation (8):

$$\text{EE/O} = \frac{P \times (t/60) \times 3.785}{V \times \log(c_0/c_f)} \tag{8}$$

where $P$ is the light source power, W; $t$ is the irradiation time, min; $V$ is the wastewater volume, L; $c_0$ is the initial concentration of AR 26, mg/L; and $c_f$ is the final concentration of AR 26, mg/L.

## 4. Conclusions

In this work, a novel UV-LED/TiO$_2$ photocatalytic system was used to simultaneously achieve UV-LED cooling and wastewater degradation. The cooling effect of LED and the decolorization efficiency of AR 26 were explored step-by-step. Additionally, a comparison of the synergistic system with a conventional photocatalytic reactor was also performed. The main conclusions are listed below:

(1) Wastewater temperature directly influences the performance of the UV-LED lamp. Hence, a high flow rate (80 mL/min) and a low temperature (20 °C) of sewage helps to ensure the long-term operational stability of LED beads.

(2) The parameters affecting the degradation rate, such as the initial concentration, TiO$_2$ concentration, and pH value, were investigated. Under an initial concentration of AR 26 of 45 mg/L, TiO$_2$ with moderate dosing (0.75 g/L) under strong acid conditions (pH = 2) helped to further improve photocatalytic activity. Under these conditions, the decolorization rate of AR 26 was more than 80%.

(3) For the same volume of treated wastewater, the degradation efficiency and photometric efficiency of the coupled system for AR 26 were 2.1 times and 1.5 times those of the conventional photocatalytic reactor, respectively. The unit power consumption of the coupled system was only 193.90 kW·h, which was 18% that of the conventional photocatalytic reactor.

(4) The novel UV-LED cooling and TiO$_2$ photocatalytic wastewater synergistic system has the advantages of simple structure, easy operation and low energy consumption. It can achieve excellent heat dissipation for high-power UV-LED and high wastewater degradation ability at the same time. It promises to be an alternative solution for treating wastewater treatment.

**Author Contributions:** H.B. and C.W. designed and conducted the experiments; X.K. compiled and analyzed the output data; H.B. designed and wrote the first version of the manuscript; C.W. and X.K. conceived and supervised the project and managed the funding acquisition. All authors have read and agreed to the published version of the manuscript.

**Funding:** This work is supported by the National Natural Science Foundation of China for Youths (21908208, 21908207), China Postdoctoral Science Foundation (2020M670659), Scientific and Technological Innovation Programs of Higher Education Institutions in Shanxi (2019L0575).

**Data Availability Statement:** The data presented in this study are available on request from the corresponding author.

**Conflicts of Interest:** The authors declare no conflict of interest.

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
