# Peer review of "Optimization Study on Synergistic System of Photocatalytic Degradation of AR 26 and UV-LED Heat Dissipation"

_catalysts, doi:10.3390/catal13040669_

Round 1

Reviewer 1 Report

My comments are as below:

-        The obtained results as shown in Fig.8. 9, 11, 13, 14… are shown with error bar. But what is replicate?

-        t1/2 is mentioned in Fig 12. It should be discussed the obtained t1/2 as mentioned in Fig 12. And explain the importance of concerning value t1/2.

-        There is a fluctuation on the degradation of AR26 at pH2, 4, 6 (<pHpzc). Please discuss.

-        It seems to be that the discussion is quite long, please concentrate and revise to be clearer.

-        The Performance comparison with conventional photocatalytic reactor section, two difference type of photocatalytic reactors (in dimension, light source…) are used. Different light sources lead to get different flux and wavelength penetrated the AR26 solution, hence getting different degradation yield. Please justify about this issue.

Reviewer 2 Report

In this manuscript, entitled “Optimization Study on Synergistic System of Photocatalytic Degradation of AR 26 and UV-LED Heat Dissipation”, Xue Kang et al. reported the UV-LED/TiO2 photocatalytic synergistic system to solve the heat dissipation problem of high-power LEDs in photocatalytic water treatment to degradation of AR 26. The authors claimed that the LED junction temperature could be controlled at 46.4°C and the degradation rate of AR 26 could reach more than 80% after 150 min at the wastewater flow rate of 80 mL per minutes, the inlet temperature of 20°C, the initial concentration of AR 26 at 45 mg/L, the concentration of TiO2 at 0.75 g/L and the pH value of 2. The designed photocatalysis system by optimization of varied parameters includes flow rate, wastewater temperature, initial AR 26 concentration, TiO2 dosage, pH value, et al. In my opinion, the results reported in this manuscript are worthy of attention. Therefore, I suggest publishing it in Catalysts after major revisions. The main recommendations that must be considered are as follows:

1. On page 2, the authors claimed that “the performance of UV-LED/TiO2 photocatalytic synergistic system was compared with the conventional photocatalytic reactor”, I think more quantitative analysis and data should be provided here, because this manuscript only discussed the decolorization effect and kinetic fit between the UV-LED/TiO2 and conventional photocatalytic system in Figure 15 and Figure 16.

2. The simulation part on page 11 (Figure 10) is very simple. I recommend the authors to provide the detailed parameter for the computer simulation, and discuss how to optimize the different conditions to improve the photocatalytic process.

3. For TiO2-based photocatalytic system, the average particle size also influences the degradation efficiency. In the manuscript, the diameter of TiO2 was 30 nm, whether the different size of TiO2 with the same photocatalytic performance?

4. In actual wastewater degradation systems, the temperature of sewage maybe difficult to regulate (hundreds of tons or even more). In addition, the sewage must contain different kinds of pollutants. The processing capability and durability of the UV-LED/TiO2 photocatalytic synergistic system should be discussed.

5. Thoroughly, check the manuscript for grammatical errors and typos. Especially, (i) the intolerable mistakes in many Figures and Tables, e.g., in Table 1, cooling water flow rate (ml/min), it should be mL/min; in Figure 5, 20ml/min should be 20 mL/min; in Figure 6, 80°Ccooling should be 80°C cooling; in Figure 7, 0°Ccooling should be 0°C cooling; etc. Once again, please double check these mistakes. (ii) Additionally, it is difficult to read the Figure 8, Figure 13, et al. please provide pictures with much higher resolution if possible. (iii) And the authors must format the references in the same style.

Round 2

Reviewer 2 Report

The revised manuscript answered all my questions, I think it could be published in Catalysts.